# RNA-Binding Proteins in Acute Leukemias

**DOI:** 10.3390/ijms21103409

**Published:** 2020-05-12

**Authors:** Konstantin Schuschel, Matthias Helwig, Stefan Hüttelmaier, Dirk Heckl, Jan-Henning Klusmann, Jessica I Hoell

**Affiliations:** 1Department of Pediatrics 1, Martin Luther University Halle-Wittenberg, 06120 Halle, Germany; konstantin.schuschel@uk-halle.de (K.S.); matthias.helwig@uk-halle.de (M.H.); dirk.heckl@medizin.uni-halle.de (D.H.); jan-henning.klusmann@uk-halle.de (J.-H.K.); 2Institute of Molecular Medicine, Martin Luther University Halle-Wittenberg, 06120 Halle, Germany; stefan.huettelmaier@medizin.uni-halle.de

**Keywords:** AML, ALL, hematopoiesis, RNA-binding protein, MSI2, IGF2BP, RBM15-MKL1, hnRNP K, nucleolin, ZFP36, WT1, CPSF6

## Abstract

Acute leukemias are genetic diseases caused by translocations or mutations, which dysregulate hematopoiesis towards malignant transformation. However, the molecular mode of action is highly versatile and ranges from direct transcriptional to post-transcriptional control, which includes RNA-binding proteins (RBPs) as crucial regulators of cell fate. RBPs coordinate RNA dynamics, including subcellular localization, translational efficiency and metabolism, by binding to their target messenger RNAs (mRNAs), thereby controlling the expression of the encoded proteins. In view of the growing interest in these regulators, this review summarizes recent research regarding the most influential RBPs relevant in acute leukemias in particular. The reported RBPs, either dysregulated or as components of fusion proteins, are described with respect to their functional domains, the pathways they affect, and clinical aspects associated with their dysregulation or altered functions.

## 1. Introduction

Cell fate decisions are directed by a complex regulatory machinery mediating the interpretation of the genomic information. Regulatory mechanisms range from epigenetic modifications, post-transcriptional modifications of the RNA to post-translational modifications of the proteome. Disturbance of this precise machinery results in aberrant cell behavior, which may cause malignant transformation: cancer.

Acute lymphoblastic leukemia (ALL) and acute myeloid leukemia (AML) are the most common and aggressive malignancies of the hematopoietic system. They both represent genetically complex and heterogeneous diseases characterized by a large number of subtypes [1,2]. Although knowledge about the molecular basis of both ALL and AML is ever expanding, overall survival of AML patients has only improved slightly during recent years [3,4]. It is believed that relapses are derived from a rare population of leukemic stem cells (LSC) that escape conventional chemotherapy and are characterized by a pronounced self-renewal capacity [5,6]. Survival rates of children diagnosed with ALL and treated with intense multi-agent chemotherapies are excellent, albeit at the price of acute and long-term toxicities and thus a reduced quality of life [7]. In both ALL and AML, targeted therapies with smaller side-effects are thus urgently needed.

While changes in the genetic and epigenetic mechanisms of leukemia development in both AML and ALL are being intensely studied, the question of how the post-transcriptional regulation of messenger RNA (mRNA)/protein expression affects the progression of leukemia has not yet been sufficiently defined.

One important step of gene regulation is the fine-tuning on the posttranscriptional level. The RNA transcript is subject to intense processing leading to the maturation of the mRNA. All these processes are strongly regulated [8]. The fate of the mature mRNA is influenced by non-coding RNAs (e.g., miRNAs) and RNA binding proteins (RBPs) as key determinants of post-transcriptional control [9]. RBPs are defined as proteins capable of binding double or single-stranded RNA and thereby participate in forming ribonucleoprotein (RNP) complexes to influence the RNA fate [10]. They influence every aspect of posttranscriptional regulations, such as (alternative) splicing, RNA modification, nuclear export, localization, stability and translation rates [11]. 

The aforementioned functions of RBPs are highly dependent on their structural features, meaning the presence and arrangement of the RNA binding domains [10]. The majority of RBPs are built from relatively few types of RNA binding domains, functionally driven by the cooperation of these domains [12]. The RNA binding domains, which are functionally important components of the RBPs described in this review, have been examined in detail in various reviews. The most common domains are in the RNA recognition motif (RRM), which binds single stranded RNA by aromatic stacking interactions (Figure 1) [13,14], the hnRNP-K homology domain (KH, Figure 1) [15] and the zinc finger domain (ZF, Figure 1) [16]. Having these versatile functions in shaping the transcriptome and proteome, RBPs have become an important class of gene expression regulators in cancer [17,18]. They contribute to oncogenesis by both dysregulation and mutation. For example, in AML, a vast number of translocations leading to fusion proteins has been described [19,20,21,22,23,24]. Mutations in proteins involved in RNA processing and metabolism have been frequently found and functionally characterized in hematological diseases [25,26,27,28,29]. In particular, multiple genetic alterations of RBPs with functions in splicing (e.g., Splicing factor 3b subunit 1 (SF3B1), U2 small nuclear RNA auxiliary factor 1 (U2AF1), serine and arginine rich splicing factor 2 (SRSF2)) have been described, especially in myeloid malignancies, and have been recently reviewed extensively [30,31,32,33,34,35], directing the focus of this review towards the wide range of RBPs regulating RNA beyond splicing. Additionally, there is a growing interest in m^6^A mRNA modifications in acute leukemias [35,36,37,38,39,40,41]. The aim of this review therefore is to highlight known disease relevant RBPs, which are playing critical roles in the development of leukemia. We focus primarily on structural aspects, their function and interplay with different pathways, the pathogenic effects (Table 1) of their dysregulation and resulting clinical prognosis plus therapeutic aspects.

## 2. IGF2BP Family

The insulin-like growth factor 2 mRNA binding proteins (IGF2BPs) are a family of well-conserved mRNA-binding proteins consisting of the three members IGF2BP1, IGF2BP2 and IGF2BP3 with 56% amino acid (AA) sequence homology. All IGF2BPs carry two RRMs at the N-terminal and four KH domains at the C-terminus (Figure 2) [45,46]. Mediated by these KH domains, two molecules of IGF2BP1 bind the target mRNA in a sequential, cooperative mechanism [47]. All three family members play important roles during fetal development in cell polarization, migration, morphology, metabolism, proliferation and differentiation [45] and are recurrently dysregulated in many different malignancies, with IGF2BP1 and IGF2BP3 being most affected [45,48]. Physiologically, there is no significant expression of either of the two proteins in hematopoietic tissue of healthy adult donors, neither in bone marrow nor in peripheral blood (an exception to this is only a low-level expression of IGF2BP3 in CD19+ cells) [49]. IGF2BPs are described as major regulators of stem cell fate. Their main cellular function is stabilization of bound mRNAs [50,51].

IGF2BP1 leads to increased proliferation and tumorigenesis because leukemia cell lines, with low expression of IGF2BP1, have a lower ability to form colonies and initiate tumors [52]. Cell lines with high expression of IGF2BP1 are AML-, chronic myeloid leukemia (CML)- and common acute lymphoblastic leukemia (cALL) cell lines. In AML, knockdown of IGF2BP1 leads to less colony-forming and higher drug-sensitivity to doxorubicin, cytarabine and cyclophosphamide [52]. IGF2BP1 binds to mRNAs of aldehyde dehydrogenase 1 member A1 (ALDH1A1), Homeobox B4 (HOXB4) and MYB proto-oncogene (MYB) and leads to higher expression of those targets [52], which are already described as dysregulated in AML [53,54,55]. Forced expression of ALDH1A1 or HOXB4 recovers the ability to form colonies in cALL and AML cell lines [52].

IGF2BP1 is also a target of the small molecule inhibitor BTYNB. BTYNB disrupts the ability of IGF2BP1 to bind to c-Myc proto-oncogene protein (MYC) mRNA. It represents an allosteric inhibitor of IGF2BP1 in melanoma and ovarian cancer cell lines and also reduces mRNA levels of other cancer-related IGF2BP1 targets like eukaryotic elongation factor 2 (eEF2), cell division cycle 34 (CDC34), calmodulin 1 (CALM1), beta-transducin repeat containing E3 ubiquitin protein ligase 1 (β-TRCP1), and collagen type V alpha 1 chain (COL5A1). In addition, there is BTYNB dependent inhibition of cell proliferation, which is not observed in IGF2BP1 negative cells [56]. BTYNB has so far not been assessed in leukemias.

IGF2BP1 and IGF2BP3 are overexpressed in translocation-ETS-leukemia virus 6 (ETV6)/runt-related transcription factor 1 (RUNX1)-positive B-ALL. The overexpression of IGF2BP1 appears specific for t(12;21)(p13;q22)-positive B-ALL [52], as it was not found in other B-ALL entities or healthy donors [49]. Mechanistically, it was shown that IGF2BP1 binds to and regulates the ETV6/RUNX1 mRNA transcript (Figure 3) [57] but also potentiates the entire ETV6/RUNX1-Ras-related C3 botulinum toxin substrate 1 (RAC1)-signal transducer and activator of transcription 3 (STAT3) signaling axis [58]. In addition, ETV6/RUNX1 appears to be the only leukemic disease in which there are almost equal levels of expression of all IGF2BPs [49]. Furthermore, IGF2BP1 binds STAT3 mRNA directly and increases its expression [58]. STAT3 is an transcription factor that regulates multiple genes and can act primarily as an oncogene, but also as a tumor suppressor [59]. Apart from ETV6/RUX1-positive leukemias, another group reported a novel translocation (t(14;17)(q32;q21)) leading to an immunoglobulin heavy locus (IGH)-IGF2BP1 fusion [60]. 

IGF2BP2 is expressed in both different hematopoietic cells as well as several subtypes of ALL. Its expression is upregulated in mixed lineage leukemia (MLL)-rearranged ALL and downregulated in transcription factor E2-alpha (E2A)/pre-B-cell leukemia transcription factor 1 (PBX1)-positive leukemias [49]. In AML, protein expression is increased up to 8-fold compared to the blood of healthy donors and IGF2BP2-knockdown showed a reduction in growth in cell culture experiments [61]. Surprisingly, genetic events targeting IGF2BP2 (such as gene amplification, deletion or mutation) are very rare. However, comparison between AML patients with low and high IGF2BP2 expression showed that patients with high expression had a lower overall survival than patients with low expression [61].

IGF2BP3 did not only show increased expression in MLL-positive B-ALLs [49] but a treatment with I-BET151, which has been described as effective treatment for MLL-positive leukemias [62], showed a dose-dependent reduction of IGF2BP3 as well as a stagnation of the leukemic cells in the G1-S phase [63]. IGF2BP3 enhances malignant properties by increasing proliferation rate and correlates with leukocyte concentration. In addition, the number of progenitor cells, which also have a survival advantage, increases upon IGF2BP3 overexpression [63]. In summary, IGF2BP3 was suggested to play a critical role in MLL-rearranged B-ALL [63]. The RNA interactome of IGF2BP3 was uncovered by individual-nucleotide resolution UV crosslinking and immunoprecipitation (iCLIP). Targeted transcripts include—among several oncogenic targets—cyclin dependent kinase 6 (CDK6) and MYC, which by themselves are involved in hematopoietic and other diseases [64,65,66,67,68]. Both targets have also been confirmed in in vivo experiments.

## 3. MSI2

The Musashi (MSI) protein family (more specifically MSI1) was initially discovered to mediate asymmetrical cell division during bristle development in Drosophila. Loss-of-function mutations result in the “double-bristle”-phenotype. The association with Musashi, an ancient Japanese samurai, who cultivated a two-sword-fighting style, led to the name assignment [71,72]. In mammals, two gene homologues have been described, MSI1 and MSI2. MSI1 binds consensus motifs in the 3′ untranslated regions of mRNAs, interacts with the poly(A)–binding protein and competes for eukaryotic initiation factor-4G. Doing so, it disturbs the translation initiation of target mRNAs [73]. For instance, MSI1 suppresses mRNA translation of NUMB endocytic adaptor protein (NUMB), a negative regulator of Notch, and therefore influences neural development [74]. It also targets other transcription factors and factors involved in the cell cycle in different malignancies [75,76]. MSI1 is a regulator of cancer stem cell functions by influencing RNA turnover [51]. MSI1 already represents a potential therapeutic target for the treatment of glioblastoma, as luteolin has been shown to inhibit the RNA-binding properties of MSI1 and disrupts cancer phenotypes in glioblastomas [77].

The Musashi-2 Protein (MSI2) is not only highly expressed in AML, but was also in one case found in a fusion protein derived from the translocation der(10)t(10;17)(q26.3;q22) [21]. The breakpoint of the translocation on chromosome 10 is within the tetratricopeptide repeat domain 40 (TTC40), which contains 58 exons. The MSI2 gene is fused to the TTC40 gene with breaks in introns 4 and 50, respectively.

Recent studies have fueled interest in MSI2, as it is expressed in hematopoietic cells [78,79], where it has a function distinct from MSI1 [78,80]. It belongs to the class A/B heterogeneous nuclear ribonucleoproteins (hnRNPs) and consists of two N-terminal RRMs (Figure 2), mediating the binding to their target mRNAs [81]. Abundance of MSI2 is higher in most primitive cells, like long-term hematopoietic stem cells (LT-HSCs), short-term hematopoietic stem cells (ST-HSCs) and multipotent progenitors (MPPs), compared to progenitor cells, differentiated myeloid cells and lymphocytes [78]. MSI2-knockout in Lin-Sca1+Kit- (LSK) cells results in a reduced engraftment upon transplantation in mice, while MSI2-overexpression in doxycycline-inducible MSI2 transgenic mice yielded an increased number of LSK cells. These findings primarily affected ST-HSCs and MPPs, whereas LT-HSCs were only mildly affected [78]. Another study also observed an enhanced long-term engraftment upon retroviral transduced overexpression of MSI2 in primary bone marrow cells and subsequent transplantation [79]. These disparities could be due to differences in the utilized methods for MSI2 overexpression, but both studies highlight the importance of MSI2.

MSI2 is dysregulated in both chronic and acute hematopoietic malignancies; MSI2 expression is increased in blast crisis CML compared to chronic-phase CML [78]. In addition, AML and ALL patients with increased MSI2 mRNA expression showed worse clinical outcome in a multitude of studies [78,82,83,84,85,86]. Thus, MSI2 can serve as a prognostic marker and possible target for therapy.

MSI2 regulates different signaling pathways, including Notch signaling through inhibition of NUMB mRNA translation (Figure 3) [78,87]. NUMB itself is a tumor suppressor and regulates Notch, Hedgehog and tumor protein P53 (p53) signaling [88]. Another pathway modulated by MSI2 is the mitogen-activated protein kinase (MAPK) pathway, which drives proliferation and suppresses apoptosis [89]. Knockdown of MSI2 caused cell cycle arrest in leukemic cell lines by increasing the expression of cyclin-dependent kinase inhibitor 1 (p21) and decreasing the expression of cyclin D1 [90] and cyclin dependent kinase 2 (CDK2) [91]. Apoptosis increased via downregulation of B-cell CLL/Lymphoma 2 (Bcl-2) and upregulation of BCL2 associated X protein (Bax) expression [90,91]. MSI2 knockdown led to the inhibition of the extracellular signal-regulated kinase (ERK)/MAPK and p38/MAPK pathways [91]. Concerning the impact on phosphorylated RAC-alpha serine/threonine-protein kinase (p-AKT) the indications diverge. While Han et al. described enhanced phosphorylation of AKT by MIS2-silencing [90], Zhang et al. did not detect differences in p-AKT [91]. The phosphoinositide 3-kinases (PI3K)/AKT pathway plays a central role in leukemic cell proliferation, growth and survival [92] and enhanced AKT activation is an important mechanism of the transformation in AML [93].

From a therapeutic point of view, MSI2 certainly is a promising target. Besides its effects on leukemia progression, an enhanced chemosensitivity to daunorubicin has been demonstrated upon MSI2 silencing [90]. Similar connections between chemoresistance and MSI2 have been shown in liver cancer stem cells and pancreatic cancer [94,95]. As a first step towards advanced clinical utility of MSI2 inhibitors, a recent study characterized the small molecule Ro 08-2750 to be a selective inhibitor of MSI2, competing with RNA binding at the RRM1. More precisely Ro 08-2750 docks between F66 and R100, stabilized by stacking interactions [96]. The homologue residues in MSI1 are F65 (labeled in Figure 1) and R99. Upon in vivo administration of this component, Minuesa et al. demonstrated a significant decrease of c-MYC levels without overt toxicity in an aggressive MLL-AF9 leukemia model, leading to a reduced disease burden.

Taken together, the combined influence on disease progression and prognosis, chemosensitivity and the concept of MSI2 as a new therapeutic target merge to a solid interest in this RBP.

## 4. HnRNP K

HnRNP K contains three KH domains, which are relevant for the binding to single stranded DNA and RNA [97,98,99,100,101,102,103], as well as an N-terminal bipartite nuclear-localization signal (NLS), an hnRNP K-specific nuclear shuttling signal (KNS), which confers the capacity for bi-directional transport across the nuclear envelope and a K protein interactive (KI) domain, responsible for the interaction between hnRNP K and other proteins in the nucleus and cytoplasm (Figure 2) [104,105,106,107,108,109]. It is a putative oncogene (reviewed in [110]) in several solid malignancies, where its overexpression is associated with poor prognosis [111,112]. In solid malignancies, there is a correlation between increased hnRNP K expression and drug resistance [113]. Similarly, AML patients with therapy-resistant disease had increased expression levels of hnRNP K. The same study showed that hnRNP K promoted survival of leukemic cells through influencing autophagy [85].

Yet, expression of hnRNP K seems to be more frequently downregulated (judging by the number of reports) in hemato-oncological diseases, which suggests that this protein can act both as an oncogene and as a tumor suppressor [114]. This dualism is based on the multifunctionality of the protein. Due to its special localization it acts in many different ways in the cytoplasm as well as in the nucleus [115]. For example, it interacts directly with other proteins, like the chromatin remodeling enzyme histone methyltransferase and Y-box binding protein 1 (YBX1) [116], binds DNA sequence specifically to the promoters of the SRC proto-oncogene (c-Src) [117] and c-myc [118] and enhances translation of c-myc mRNA by binding to the internal ribosomal entry site (IRES) [119]. On the other hand, hnRNP K also silences the translation of 15 lipoxygenase (LOX) mRNA by binding to its differentiation control element (DICE) located in the 3′UTR [120]. hnRNP K is also capable of influencing alternative splicing directly [121] and indirectly by upregulating splicing factors [122].

In AML, hnRNP K binds to the mRNA of the transcription factor PU.1 (PU.1), which regulates cell differentiation and maturation (Figure 3) [123,124]. In cooperation with other transcription factors like GATA-binding factor 1, (GATA-1), GATA-binding factor 2 (GATA-2) and RUNX-1, it modulates multiple genes coding cell surface proteins, cytokines and their respective receptors [123]. HnRNP K causes a reduction in PU.1 levels and vice versa [125]. The significance of hnRNP K was also investigated in vivo. While biallelic loss of hnRNP K led to embryonic lethality, haploinsufficent mice were significantly smaller than wild-type mice and had decreased overall survival [126]. Since hnRNP K was already described to be associated with the p53/p21 pathway [127,128], Hnrnpk^+/-^-Mouse embryo fibroblasts (MEF) were created. They had less transcriptional activation of p21. This could be confirmed by reduced expression of p21 in bone and liver tissue of haploinsufficient mice. Its gene is, according to chromatin immunoprecipitation assays (ChIP), the binding partner of hnRNP K [126]. Furthermore, hnRNP K binds to the mRNA of p21 [129] and to MYC mRNA [130], which is described to inhibit p53 in leukemic cells [131]. The stability of hnRNP K and its influence on p21 is increased when hnRNP K is conjugated with a small ubiquitin-related modifier (SUMO) at Lys422. This conjugation is for example facilitated by the E3 SUMO-protein ligase protein inhibitor of activated STAT 3 (PIAS3) upon UV damage and can be reversed by the Sentrin-specific protease 2 (SENP2) [132]. Moreover, haploinsufficient mice had an increased serum concentration of phos-STAT3 and TNF-α compared to wild-type mice. Reduced expression of hnRNP K led to a significant increase of myeloid cells including neutrophils, basophilic granulocytes and platelets, as well as increased CD11b/LyG6 double positive cells. Moreover, serum levels of interleukin (IL)-3 and IL-6, which promote the proliferation of myeloid cells, were increased. In addition, hnRNP K haploinsufficient mice were more likely to develop hematological malignancies [126].

In humans, hnRNP K was significantly reduced in CD34^+^ cells of AML patients compared to CD34+ cells from healthy donors [126]. 

All in all, hnRNP K has properties both of an oncogene and tumor suppressor. Yet, in the context of acute leukemia, in particular AML, it mostly behaves like a tumor suppressor.

## 5. Nucleolin

Nucleolin (NCL), which has 4 RRM-domains (Figure 2) [133], is mainly found in the nucleolus, where it is also the most highly expressed nucleolar protein. It associates with ribosomal DNA (rDNA) and is involved in ribosomal synthesis as being involved in C/D box Pre-rRNP and rRNA complexes, which are essential for the clonogenicity of leukemic blasts [134]. It also impacts DNA-methylation [135,136]. NCL overexpression is associated with many malignant diseases including gastric cancer [137], breast cancer [138] and CML [139]. 

In acute leukemias, NCL is particularly relevant in AML, where its overexpression correlates with poor prognosis, especially in elderly patients (>60 years of age) [140,141]. In AML cell lines, knockdown of NCL leads to a decreased proliferation rate caused by a higher apoptotic rate through activation of the caspase pathway. This was confirmed in vivo, as mice transplanted with NCL-inactivated leukemic cells had a significantly lower leukemic volume compared to controls 24 days after injection [140].

Overexpression of NCL increased phosphorylation of nuclear factor kappa B (NFκB), associated with the overexpression of DNA methyltransferase (DNMT) 1 (and DNMT3a and DNMT3b) and resulting DNA hypermethylation. Drug-induced (AS1411) inactivation of NCL and NFκB led to hypomethylation, activation of caspase signaling and thus to reduced colony growth, as well as to a reactivation of the tumor suppressor p15^INK4B^ [140]. In addition to the NFkB/DNMT axis, overexpression of NCL also suppresses p53 translation and induction after DNA damage by binding to its 5′UTR [142]. Furthermore, it binds to the 3′UTRof BCL2, leading to its mRNA stabilization and subsequent anti-apoptotic effects (Figure 3) [143,144,145,146,147].

A mutation of the interaction partner nucleophosmin (NPM), a highly prominent mutation in normal karyotype AML [148], results in the loss of interaction between NCL and NPM in vitro [149]. NPM mutations mostly lead to cytoplasmic mislocalization of the protein, but even after drug-induced re-localization of NPM into the nucleolus there is no interaction between NCL and NPM [149]. NCL expression is not dependent on the presence of the NPM mutation [149]. It is also interesting to note that all-trans retinoid acid (ATRA), already described in relation to NPM [150,151], leads to a shift into G0/G1 phase and also a relocation of NPM, even if not restoring the interaction between NCL and NPM [149]. NCL seems to be an interesting target and is already proven to be responsive to oridonin, a direct inhibitor of NCL, potentially impairing its ability to stabilize specific mRNAs [152].

## 6. WT1

Originally described in association with Wilms tumor [153] (a kidney tumor), Wilms Tumor 1 (WT1), which plays a key role in urogenital development [154,155,156], is now more generally described as an oncogene in some cancer entities including leukemias [157] and also as tumor suppressor in others [158,159,160,161,162]. 

Its oncogenic function in AML is supported by a meta-analysis showing a correlation of WT1 overexpression in AML and a reduced overall and disease-free survival [163]. Additionally, WT1 overexpression in peripheral blood after stem cell transplantation is predictive of relapse in AML [164] and a higher risk of mortality following stem cell transplantation [165]. Relapse could be prevented in high-risk AML by T-cell receptor therapy targeting WT1 [166] but due to a small number of patients enrolled in this study, this needs further investigation.

WT1 binds both DNA and RNA mainly through Zink-finger-domains (Figure 2) (reviewed in [156]) and has two isoform (KTS(+) and KTS(-)), which differ in the so-called KTS insert, which is present or absent following alternative splicing events [167]. Both isoforms have different binding specificities and regulate transcription depending on the isoform [168]. WT1 KTS (-) acts as an oncogene by inducing expression of the transcription factor BCL2 associated athanogene 3 (BAG3) [169], which interacts with 70 kD heat shock protein (HSP70) and sustains NFκB [170], thereby promoting an antiapoptotic effect [171].

Mutations of WT1 are mutually exclusive with mutation in either ten-eleven translocation methylcytosine dioxygenase 2 (TET2), isocitrate dehydrogenase 1 (IDH1), isocitrate dehydrogenase 2 (IDH2) or CCAAT enhancer binding protein alpha (CEBPA) [172,173]. The latter two are linked with DNA hypermethylation in AML [174,175]. TET2 co-activates WT1 targeted genes through binding the transcription start sites (TSSs) and CpG islands (CGIs) of WT1-target genes and to the WT1 zinkfinger domain itself. Co-overexpression of WT1 and TET2 leads to a dose-dependent overexpression of WT1 targets, including genes of the Wnt, MAPK and axon guidance pathway. AML cell lines have a reduced proliferation rate and clonogenicity when TET2 is overexpressed. Those effects dissappear if WT1 is knocked down [173]. In WT1 or TET2 mutated AML cell lines, the lncRNA maternally expressed gene 3 (MEG3), which reduces cell proliferation and induces GO/G1-phase and apoptosis by influencing p53 expression [176], is significantly downregulated. WT1, with TET2 as cofactor, transcriptionally regulates MEG3 expression (Figure 3) [177]. 

WT1-haploinsufficient mice show altered myeloid differentiation and leukemic transformation, whereas WT1 null mice do not [178]. Together with Flt3-ITD mutations, which are associated with poor prognosis in AML [179], WT1-haploinsufficiency (WT1^+/R394W^) induces fully penetrant AML in mice [178]. WT1^+/R394W^ alone leads to myelodysplastic syndrome and increased myeloid progenitors [180].

Besides its oncogenic function, WT1 increases fas-receptor ligand (CD95L) expression [181], which is involved in activation-induced cell death (AICD) in B- and T-lymphocytes [182] through binding the EGR promotor. The KTS(-) isoform more strongly enhances CD95L [181], thus potentiating its tumor suppressive ability.

All in all, WT1 seems mostly relevant as a transcription factor in the context of acute leukemias; further investigation is needed to clarify its role as RBP in these diseases.

## 7. ZFP36L1/2

The zinc finger protein homolog 36 (ZFP36) family consists of four members in mammals, which are ZFP36 (TIS11, TPP, Nup475, GOS24), ZFP36L1 (TIS11b, Berg36, ERF-1, BRF-1), ZFP36L2 (TIS11d, ERF-2, BRF-2) and ZFP36L3 [183]. ZFP36L3 is described only in rodents and was not detected in human tissues [184].

The members of this protein family are characterized by two tandemly repeated zinc finger motifs (Figure 2) of the Cys-Cys-Cys-His (CCCH) type, through which they bind to adenine uridine (AU) rich elements (AREs). ARE binding mediates mRNA decay [185], by binding to 11-mers consisting of UUAUUUAUUUA or AUUUAUUUAUU with tolerated UU-interspacers as the preferred binding sequence [186].

ZFP36L1 and ZFP36L2 play important roles in hematopoiesis. Zfp36l2 knockout mice die within two weeks after birth due to defective hematopoiesis, leading to intestinal or other hemorrhage [187]. 

A confirmed target of both ZFP36L1 and ZFP36L2 is notch receptor 1 (Notch1), whose expression is redundantly suppressed by both proteins at the posttranscriptional level. Like mice overexpressing Notch1, Zfp36l1-Zfp36l2 double knockout mice develop T-ALL. Thymus development of those mice is perturbed, leading to an accumulation of cells that have passed through the ß-selection checkpoint without prior expression of T-cell receptor β [188]. While other members of the ZFP36 family exhibit antiproliferative and apoptotic effects [189], mutations in ZFP36L2 lead to a less pronounced antiproliferative effect. Since wild-type ZFP36L2 inhibits proliferation by activating the S phase checkpoint pathway and thereby delaying progression through the S phase, the I373fsX91 frame shift mutation variant lacks these properties. This mutation leads to the loss of conserved C-terminal domains, implying their functional importance [190].

ZFP36L1 is a positive regulator of monocyte/macrophage differentiation. It is upregulated during monocyte/macrophage differentiation and negatively regulates vascular endothelial growth factor (VEGF) and BCL2 mRNA expression (Figure 3) by binding to the AREs in their 3’UTRs [191,192]. VEGF signaling promotes autocrine AML blast cell proliferation, survival and chemotherapy resistance [193], while BCL2 proteins regulate mitochondrial outer membrane permeabilization leading to the irreversible release of intermembrane space proteins and subsequent apoptosis [194]. ZFP36L1 also binds and regulates CDK6 mRNA leading to its decay [68]. The mRNA-destabilizing activity is mediated by different kinases. These phosphorylate ZFP36L1 among other residues at Ser92 and Ser203, impairing its functionality and stabilizing its mRNA targets [195,196,197]. CDK6 encodes a kinase that is involved in the regulation of the G1 phase progression and G1/S transition of the cell cycle [198]. It activates the cell cycle in the early G1 phase through interactions with cyclins D1, D2 and D3 [199]. Thus, ZFP36L1 also takes part in the cell cycle regulation.

ZFP36L1 is downregulated in AML patients and upregulated in controls [68]. It is frequently suppressed in different cancers, suggesting its utility as a prognostic indicator [200].

## 8. RBM15-MKL1

RBM15-MKL1, also known as OTT-MAL, is a chimeric fusion protein first discovered in 2001 in t(1;22)(p13;q13) rearranged cases of acute megakaryoblastic leukemia (AMKL) [201,202] and was overall found to be mainly associated with pediatric de novo AMKL, which accounts for two thirds of AMKL in children [203]. It is associated with female sex predominance, young age, and poor prognosis [204,205,206]. There is only one known adult case of AMKL with involvement of RBM15-MKL1 [207].

RNA-binding motif protein-15 (RBM15), also known as OTT (“one twenty-two”), harbors a SPEN paralogue and orthologue C-terminal (SPOC) domain and three RRMs (Figure 4) [208]. It belongs to the split end (SPEN) family of evolutionary conserved proteins involved in cell fate decisions [209]. RBM15 regulates RNA splicing of transcription factors important for megakaryocytic differentiation, such as GATA1 and RUNX1. RMB15 is essential for the development of different tissues and cell types, like the maintenance of the homeostasis of long-term HSCs and for megakaryocyte and B cell differentiation [208,210,211]. A reduction of RBM15 levels favors the formation of alternatively spliced isoforms such as RUNX1a and GATA1s [212]. GATA1s is the short form of GATA1 lacking exon2 [213], which supports unrestricted proliferation due to losing the interaction with transcription factor E2F1 (E2F1) [214] and plays a major role in transient myeloproliferative syndrome of children with trisomy 21 [215,216,217,218]. RUNX1a, also known as AML1a, is overexpressed in different types of leukemia [219]. RBM15 additionally plays important roles in the thrombopoietin response of HSCs [220] and the expressional regulation of the proto-oncogene c-Myc [210].

MKL1, also known as megakaryocytic acute leukemia (MAL), basic SAP and coiled-coil domain containing protein (BSAC) or myocardin-related transcription factor A (MRTF-A), is one of three members of the myocardin family. This family is characterized by N-terminal RPEL-repeats, a binding site for serum response factors, a leucine zipper-like domain that influences homo- and heterodimerization, a C-terminal transactivation (TA) domain, and also a SAP domain (Figure 4) [221]. Dependent on the zipper-like domain, its dimer undergoes a Crm1-mediated cytosolic export. A disturbed dimerization (e.g., by a mutated coiled-coil domain) leads to an interfered interaction with chromosomal maintenance 1 (Crm1) and therefore a monomeric protein remains in the nucleus followed by abolished cytosolic interactions [222]. MKL1 promotes transcriptional activation of serum response factor-responsive genes, like growth factors and differentiation-associated muscle-genes in different cell types [221]. Doing so, it drives expression of genes regulating the cytoskeleton during development, morphogenesis and cell migration. Besides that, the protein has an anti-apoptotic effect in embryonic fibroblasts [223].

The t(1;22) RBM15-MKL1 fusion is located within a 4-kb intron of the RBM15 gene downstream of the C-terminal SPOC domain on chromosome 1 [224] and fuses this gene with MKL1. The resulting chimeric protein is 1833 amino acids long and encompasses all putative functional motifs encoded by each gene [201]. Nevertheless, the functionality of the single components is impeded, thus altering their respective regulatory functions leading to the development of leukemia [212,224,225]. Contrary to the predominant cytoplasmatic localization of dimeric wild-type MKL1, RBM15-MKL1 is exclusively localized in the nucleus. There, MKL1 exhibits enhanced transcriptional activity to various promotors containing the YY1-binding sequences like the glycoprotein VI (GPVI) promotor [226]. Wild-type RBM15 physiologically interacts with histone deacetylase 3 (HDAC3) [226], but also with several histone methyltransferases regulating H4 acetylation and H3K4me3 marks. However, RBM15-MKL1 does not interact with HDAC3, which is mediated by the TA domain of MKL1, since deletion of the TA domain in RBM15-MKL1 leads to the reconstitution of HDAC3 interaction. This loss of suppressor function also supports the development of leukemia [226].

Beside those altered functions of the fusion protein components, the expression of the fusion protein and its endogenous components is regulated reciprocally. While RBM15 does not change the expression of MKL1 and MKL1 does not affect RBM15 expression, the chimeric protein RBM15-MKL1 decreases the RBM15 expression ~8-fold and the expression of MKL1 is increased ~3-fold. In addition, a forced expression of RBM15 decreases the expression of the fusion protein in a ~2.5-fold manner. This regulatory loop is likely based on the N-terminal domain, although the exact mechanism is not yet clear [225].

## 9. CPSF6

Another fusion protein containing a tyrosine kinase is t(8;12)(p12;q15) CPSF6-FGFR1, which is related to the 8p11 myeloproliferative syndrome [19], a very aggressive disease with a high rate of progression to an AML resistant to conventional chemotherapy with a median survival rate of 12 months [227,228,229]. The cleavage and polyadenylation specificity factor 6 (CPSF6), also known as CFIM, is a nuclear protein important for pre-mRNA processing like splicing factor proline and glutamine rich (SFPQ) and consists of heterologously expressed 25- and 68-kDa subunits, resulting in the CF I(m)68/25 heterodimer. The 68 kDa subunit has a modular domain organization consisting of an N-terminal RRM, a proline-rich central region, and a C-terminal arginine/serine-rich(RS)-like domain. The dimer binds to pre-mRNA, which is one of the earliest steps in the assembly of the cleavage and polyadenylation machinery, followed by recruitment of other processing factors [230,231,232]. The CPSF6-FGFR1 fusion protein connects the RRM of CPSF6 with the kinase domain of FGFR1 [19]. 

Another fusion protein harbouring CPSF6 is CPSF6-RARG, which was found in several hematopoietic malignancies, showing morphological and immunophenotypical features of classical hypergranular acute promyelocytic leukemia (APL) [23,24]. Retinoic acid receptor gamma (RARG) has a DNA binding domain (DBD) and a ligand-binding domain (LBD), assuming a ligand-inducible transactivation function [233] and also appears in other translocations, resulting in NUP-RARG and PML-RARG fusion transcripts. These cases show characteristics like APL [234,235]. CPSF6-RARG appears in two isoforms, including exon 1-4 of CPSF6 and either exon 1-10 or exon 4-10 of RARG. Both isoforms combine the RRM of CPSF6 and the main RARG domains of DBD and LBD [24]. Patients with this translocation are resistant to ATRA and arsenic. Conventional anthracycline and cytarabine-based chemotherapies are also ineffective [23,24,236,237], but sensitivity to homoharringtonine was reported in one case [237].

## 10. Tyrosine Kinase Fusions Involving RBPs

In ALL, many translocations have been described, prominently those involving the tyrosine kinase ABL1. For ABL1, multiple fusion partners have been described, the most prominent in Philadelphia-positive ALL certainly being BCR-ABL1 p190 [238,239,240,241]. Another example of an ABL1 fusion involving an RBP is SFPQ-ABL, which has been described as a cause of B-cell progenitor ALL. This translocation was located in exon 9 of the splicing factor SFPQ and exon 4 in ABL [19,20]. This is unusual for ABL translocations as they usually affect exon 2 [238]. The SFPQ-ABL fusion protein contains the N-terminal proline rich region and the two RRMs from SFPQ fused to the part of the SH2 domain of ABL [19]. 

Tyrosine kinase harbouring fusion proteins are also present AMKL. For instance, the fusion protein RBM6-CSF1R is expressed in the AMKL cell line MKPL-1, generated by the translocation t(3;5)(p21;q33) [22]. This fusion protein connects 36 N-terminal amino acids of RNA-binding motif 6 (RBM6) to the C-terminal 399 amino acids the colony-stimulating factor 1 receptor (CSF1R), a class III receptor tyrosine kinase. The resulting truncation of CSF1R is causative of the phenotype as RBM6 lacks its functional domains in this fusion protein. The truncation of the CSF1R kinase leads to constitutive activation, due to the loss of its regulatory juxtamembrane domain [22].

The CPSF6-FGFR1 fusion involving the tyrosine kinase FGFR1 was already described above.

## 11. Discussion

As this review illustrates, much is already known about a select number of RBPs and their role in leukemogenesis. However, the complex regulatory network of RBPs in general is still only incompletely understood. Yet, knowledge is constantly increasing and novel methods such as CRISP-Cas9 screens [242,243] for the discovery of new disease-relevant RBPs are emerging [244,245,246]. In addition, established methods are further improving [247,248] and targets of relevant RBPs are the subject of detailed investigation [249,250].

For instance, a domain-focused CRISP-Cas9 screen targeting an RNA-Binding Protein Network in AML could recently unveil novel functions of the RBP RBM39, which is crucial for maintaining RNA splicing and AML survival [244]. As a splicing factor, RBM39 is out of scope of this review, yet offers a proof-of-principle regarding the potential of CRISPR screens to identify novel pathogenic RBPs.

This research culminates in the potential of developing novel therapeutic possibilities (Table 2). Representing this potential, Ro binds to MSI2 and competes for its RNA binding in biochemical assays. Ro treatment in mouse and human myeloid leukemia cells results in increased apoptosis and differentiation, while inhibiting known MSI-targets [96]. Since MSI2 has an impact on the sensitivity to daunorubicin [90] and other RBPs also influence the chemosensitivity of certain cancers [94,113,251,252,253], targeting those RBPs could potentially synergize with existing therapies.

Besides the Ro-mediated inhibition of MSI2, other small molecule drugs targeting RBPs with potential use for leukemia treatment emerge [254]. The bioactive plant diterpene oridonin for example exerts anti-leukemic effects by inhibiting the activation of mTOR/P70/4EBP1, Raf/ERK and STATS signaling pathways, while down-regulating the expression of Bcl-2 and up-regulating the expression of BAX [255]. It is also a direct inhibitor of NCL, potentially impairing its ability to stabilize specific mRNA [152]. Yet, for most of the described RBPs in this review, no drugs are currently being tested, let alone clinically available.

In summary, this review underscores the influence of RBPs on acute leukemias and thus the importance of understanding their interaction networks. Future studies will further contribute to this knowledge and will hopefully enable the discovery of new targeted therapies.

## Figures and Tables

**Figure 1 ijms-21-03409-f001:**
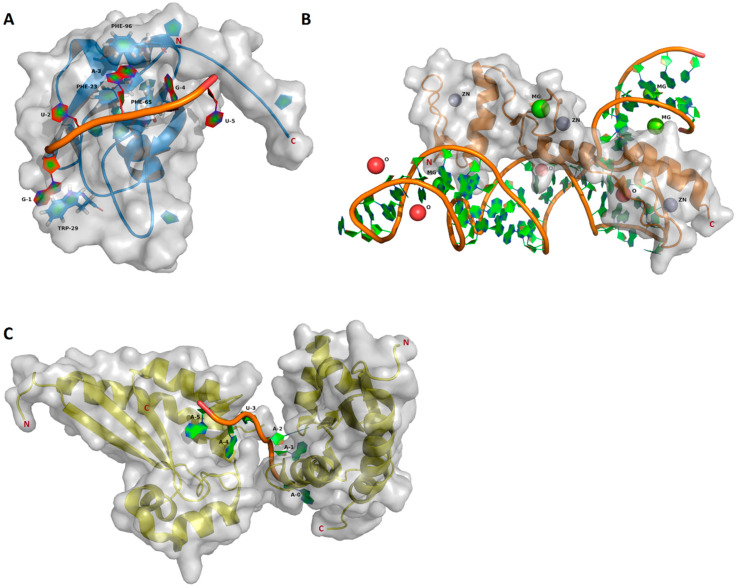
Structures of the RNA binding domains, visualized with PyMOL version 1.5.0.4 using the corresponding RCSB protein data bank entries. Protein backbones are represented as blue (RNA recognition motif (RRM)), orange (Zinc finger domain (ZF)) and yellow (hnRNP-K homology (KH)) cartoon models. The solvent accessible surfaces are indicated in transparent gray. The bound RNA cartoon models are shown with filled aromatic rings. (A) Msi1 RRM1 in a complex with r(GUAGU) (NUMB Endocytic Adaptor Protein 5 (numb5)). The cartoon model contains filled aromatic rings. Protein side chains contributing to the RNA binding are shown as a stick model with filled aromatic rings (blue) (PDB ID: 2RS2) [14]. (B) ZF-RNA complex with ions. Solvent components are shown as balls: zink, grey; magnesium, green; oxygen (HOH), red (PDB ID: 1UN6) [42]. (C) Two KH domains in complex with r(AAAUAA). One KH domain binds the 5′AAA moiety and another KH binds the 3′UAA moiety (PDB ID: 5ELS) [43].

**Figure 2 ijms-21-03409-f002:**
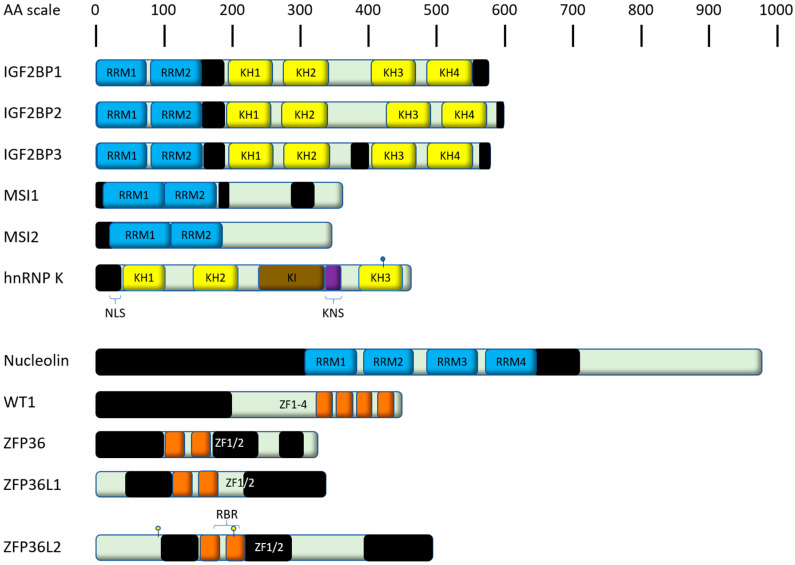
Shown are human RNA binding proteins (RBPs) (together with associated family members) playing a major role in acute leukemias as discussed in this review. Domains are color-coded and derived from their respective entry at UniProt [69], unless stated differently. Proteins are scaled to amino acid (AA) length. Black bars indicate intrinsically disordered regions predicted by IUPred2A [70]. RRM, RNA recognition motif; NLS, nuclear localization signal; KNS, hnRNP K-specific nuclear shuttling signal; KI, K protein interactive domain; KH, hnRNP K homology domain; RBR, RNA binding region; ZF, zinc finger domain. The conjugation with the small ubiquitin-related modifier (SUMO) of hnRNP K is indicated with a blue ball. Phosphorylations of ZFP36L1 discussed in this review are indicated with yellow-core balls.

**Figure 3 ijms-21-03409-f003:**
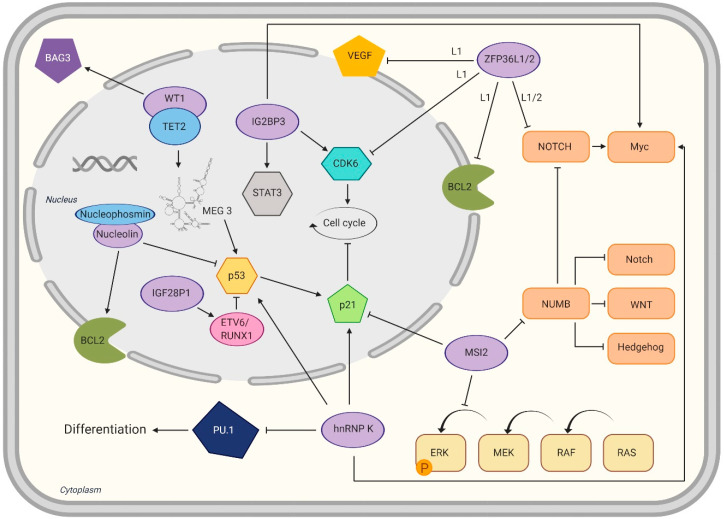
RNA binding proteins (RBPs), their RNA targets and their implications in cellular pathways. The RBPs presented in this review, which are not a component of a fusion construct, are displayed at their cellular localization: nucleus, light-grey; cytoplasm, beige. The effects of RBP regulation can be either inhibitory (

) or activating (→).

**Figure 4 ijms-21-03409-f004:**
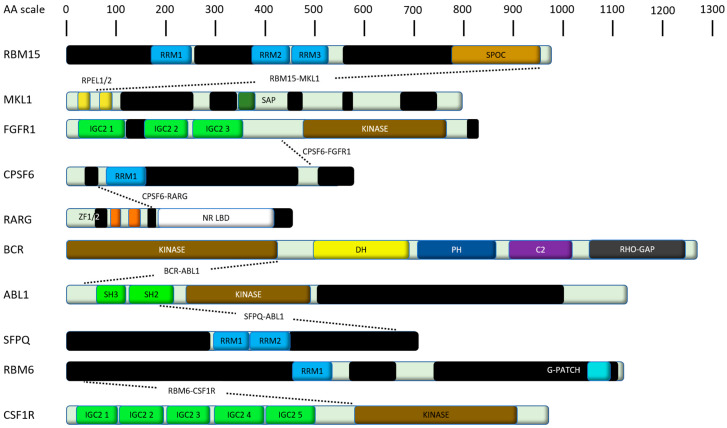
Shown are human fusion proteins playing a major role in leukemias as discussed in this review. Domains are color-coded and derived from their respective entry at UniProt [69], unless stated differently. Proteins are scaled to amino acid (AA) length. Black bars indicate intrinsically disordered regions predicted by IUPred2A [70]. RRM, RNA recognition motif; SPOC, Spen paralogue and orthologue C-terminal; RPEL, RPxxxEL repeat (where x is any amino acid); RBR, RNA binding region; ZF, Zinc finger domain; SAP, SAF-A/B Acinus and PIAS domain; IGC, Immunoglobulin constant domain; NR LBD, Nuclear receptor ligand-binding domain; DH, Dbl homology domain; PHPleckstrin homology domain, C2, Protein kinase C conserved region 2; RHO-GAP, GTPase-activator protein for Rho-like GTPases; SH, Src homology domain; G-PATCH, G-patch domain.

**Table 1 ijms-21-03409-t001:** General overview of leukemia-related RNA binding proteins (RBPs), their localization [44] and disease properties.

RBP	Affected Leukemia Type	Localization	Proliferation Apoptosis Differentiation			Clinical Prognosis (When Altered)
			P	A	D	
IGF2BP	AML/ALL	Nucleus	↑			Poor
MSI2	AML/ALL	Cytosol	↑	↓		Poor
RBM15	-	Nucleoplasm	↓		↑	n/a-
RBM15-MKL1	AML/ALL	Nucleus	↑			Poor
hnRNP K	AML	Nucleoplasm/Cytosol	↓		↑	Unknown
Nucleolin	AML	Nucleoplasm, Nucleoli	↑			Poor
ZFP36L1/2	AML/ALL	Nucleus, Cytoplasm	↓	↑		Unknown
WT1	ALL	Nucleoplasm	↑			Poor

**Table 2 ijms-21-03409-t002:** **RNA binding proteins** (RBPs) and known targeting drugs.

RBP	Targeting Drug	Mechanism of Action
IGF2BP1	BTYNB	Allosteric inhibitor of Igf2BP1 in melanoma and ovarial cancer
IGF2BP3	I-BET151	dose-dependent reduction of IGF2BP3 and stagnation of the cells in G1-S phase
MSI2	Ro 08-2750	Competitive inhibition of RNA binding at the RRM1
NCL	Oridonin	Direct inhibitor of NCL
	AS1411	Inactivation of NCL and NFκB leads to hypomethylation and activation of caspase signaling

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
