# Peer review of "RNA-Binding Proteins in Acute Leukemias"

_ijms, 2020, doi:10.3390/ijms21103409_

Round 1

Reviewer 1 Report

Possible improvements:

a) Figure 1:
- it is not clear if it is for human proteins
- MSI1, ZLP36, ZLP36L3 are not present on the diagram (you draw all four members os family IGF2BP, why did not you do the same to MSI and ZFP36 families?)
- similarly, you picture only two fusion proteins while in the text you mention many more (e.g. CPSF6/FGFR1); actually you could make two separate figures, one for standard RBPs and second for fusion proteins
- you could add information about known posttranslational modifications of those proteins which are crucial for protein function/malignance
- apart from domains, you could also mark intrinsically disordered regions (you can predict them quite confidently from sequence using numerous bioinformatics tools)

b) as you mention several times that you focus on the structure of those proteins you could add some information from PDB bank, for instance, you can make a figure with domain structure like one RRM, one KH, one ZF and so on (it would good impression to the readers how those proteins can look in crucial parts at least)

c) Figure 2
- it lacks some proteins mentioned in the text (e.g. MSI1, IGF2Bp2, nucloplastin, NFkB) - the same problem is in Table 1

- in the title, you mention cellular paths, but you actually do not show any (if the reader will not be familiar with the names and function of some of those proteins he will not deduce in what pathways those proteins are involved, you only show interacting partners)

- the cytoplasm in gray can be avoided (it blurs the image, similarly blue NOTCH and NUMBS partners)

d) separate table with the information about RBPs and drug association would be also nice (you can also mention target pocket or domain for given drug)

e) some editing, correction of style, removing white spaces or adding extra ones needed (e.g. "improved side-effect profiles -> "smaller side-effects", "[96-101].It" -> "[96-101].It")'

f) not all abbreviations are explained (e.g. iCLIP)

Reviewer 2 Report

The manuscript reviews the field of RNA-binding proteins (RBPs) as crucial regulators of cell fate and comments on the future direction of the field. The review is well written and comprehensive covering the latest advances and controversies in the field. However, the review does not offer a broad information to readers given the vastly large numbers of published papers and reviews in this field. Unfortunately the authors have discussed only a few RBPs, which they considered to be the most implicated in cancer and acute leukemias in particular.

Finally, I think that a table summarizing and concentrating the clinical and relevant aspects of therapy targeting these RBPs in acute leukemia will make the review even more interesting and facilitate its reading.
